# Contrastive Learning Using Spectral Methods

**James Zou**
Harvard University

**Daniel Hsu**
Columbia University

**David Parkes**
Harvard University

**Ryan Adams**
Harvard University

## Abstract

In many natural settings, the analysis goal is not to characterize a single data set in isolation, but rather to understand the difference between one set of observations and another. For example, given a background corpus of news articles together with writings of a particular author, one may want a topic model that explains word patterns and themes specific to the author. Another example comes from genomics, in which biological signals may be collected from different regions of a genome, and one wants a model that captures the differential statistics observed in these regions. This paper formalizes this notion of contrastive learning for mixture models, and develops spectral algorithms for inferring mixture components specific to a foreground data set when contrasted with a background data set. The method builds on recent moment-based estimators and tensor decompositions for latent variable models, and has the intuitive feature of using background data statistics to appropriately modify moments estimated from foreground data. A key advantage of the method is that the background data need only be coarsely modeled, which is important when the background is too complex, noisy, or not of interest. The method is demonstrated on applications in contrastive topic modeling and genomic sequence analysis.

## 1 Introduction

Generative latent variable models offer an intuitive way to explain data in terms of hidden structure, and are a cornerstone of exploratory data analysis. Popular examples of generative latent variable models include Latent Dirichlet Allocation (LDA) [1] and Hidden Markov Models (HMMs) [2], although the modularity of the generative approach has led to a wide range of variations. One of the challenges of using latent variable models for exploratory data analysis, however, is developing models and learning techniques that accurately reflect the intuitions of the modeler. In particular, when analyzing multiple specialized data sets, it is often the case that the most salient statistical structure—that most easily found by fitting latent variable models—is shared across all the data and does not reflect interesting specific local structure. For example, if we apply a topic model to a set of English-language scientific papers on computer science, we might hope to identify different co-occurring words within subfields such as theory, systems, graphics, *etc*. Instead, such a model will simply learn about English syntactic structure and invent topics that reflect uninteresting statistical correlations between stop words [3]. Intuitively, what we would like from such an exploratory analysis is to answer the question: *What makes these data different from other sets of data in the same broad category?*

To answer this question, we develop a new set of techniques that we refer to as *contrastive learning* methods. These methods differentiate between *foreground* and *background* data and seek to learn a latent variable model that captures statistical relationships that appear in the foreground but do not appear in the background. Revisiting the previous scientific topics example, contrastive learning could treat computer science papers as a foreground corpus and (say) English-language news articles as a background corpus. As both corpora share the same broad syntactic structure, a contrastive foreground topic model would be more likely to discover semantic relationships between words that are specific to computer science. This intuition has broad applicability in other models and domains

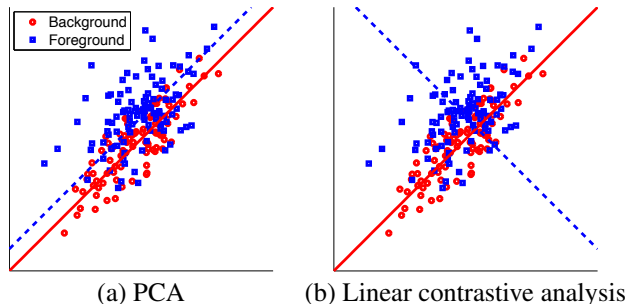

| (a) PCA | (b) Linear contrastive analysis |

Figure 1: These figures show foreground and background data from Gaussian distributions. The foreground data has greater variance in its minor direction, but the same variance in its major direction. The means are slightly different. Different projection lines are shown for different methods, to illustrate the difference between (a) the purely unsupervised variance-preserving linear projection of principal component analysis, (b) the contrastive foreground projection that captures variance that is not present in the background.

as well. For example, in genomics one might use a contrastive hidden Markov model to amplify the signal of a particular class of sequences, relative to the broader genome.

Note that the objective of contrastive learning is not to discriminate between foreground and background data, but to learn an interpretable generative model that captures the differential statistics between the two data sets. To clarify this difference, consider the difference between principal component analysis and contrastive analysis. Principal component analysis finds the linear projection that maximally preserves variance without regard to foreground versus background. A contrastive approach, however, would try to find a linear projection that maximally preserves the foreground variance that is not explained by the background. Figure 1 illustrates the differences between these. Novelty detection [4] is also related, but it does not directly learn a generative model of the novelty.

**Our contributions.** We formalize the concept of contrastive learning for mixture models and present new spectral contrast algorithms. We prove that by appropriately "subtracting" background moments from the foreground moments, our algorithms recover the model for the foreground-specific data. To achieve this, we extend recent developments in learning latent variable models with moment matching and tensor decompositions. We demonstrate the effectiveness, robustness, and scalability of our method in contrastive topic modeling and contrastive genomics.

## 2 Contrastive learning in mixture models

Many data can be naturally described by a mixture model. The general mixture model has the form

$$p(\{x_n\}_{n=1}^N; \{(\mu_j, w_j)\}_{j=1}^J) = \prod_{n=1}^N \left[ \sum_{j=1}^J w_j f(x_n | \mu_j) \right] \quad (1)$$

where $\{\mu_j\}$ are the parameters of the mixture components, $\{w_j\}$ are the mixture weights, and $f(\cdot | \mu_j)$ is the density of the $j$-th mixture component. Each $\mu_j$ is a vector in some parameter space, and a common estimation task is to infer the component parameters $\{(\mu_j, w_j)\}$ given the observed data $\{x_n\}$.

In many applications, we have two sets of observations $\{x_n^{\mathsf{f}}\}$ and $\{x_n^{\mathsf{b}}\}$, which we call the foreground data and the background data, respectively. The foreground and background are generated by two possibly overlapping sets of mixture components. More concretely, let $\{\mu_j\}_{j \in A}$, $\{\mu_j\}_{j \in B}$, and $\{\mu_j\}_{j \in C}$ be three disjoint sets of parameters, with $A$, $B$, and $C$ being three disjoint index sets. The foreground $\{x_n^{\mathsf{f}}\}$ is generated from the mixture model $\{(\mu_j, w_j^{\mathsf{f}})\}_{j \in A \cup B}$, and the background $\{x_n^{\mathsf{b}}\}$ is generated from $\{(\mu_j, w_j^{\mathsf{b}})\}_{j \in B \cup C}$.

The goal of contrastive learning is to infer the parameters $\{(\mu_j, w_j^{\mathsf{f}})\}_{j \in A}$, which we call the *foreground-specific model*. The direct approach would be to infer $\{(\mu_j, w_j^{\mathsf{f}})\}_{j \in A \cup B}$ just from $\{x_n^{\mathsf{f}}\}$, and in parallel infer $\{(\mu_j, w_j^{\mathsf{b}})\}_{j \in B \cup C}$ just from $\{x_n^{\mathsf{b}}\}$, and then pick out the components specific to the foreground. However, this involves explicitly learning a model for the background data, which

is undesirable if the background is too complex, if $\{x_n^{\mathsf{b}}\}$ is too noisy, or if we do not want to devote computational power to learn the background. In many applications, we are only interested in learning a generative model for the difference between the foreground and background, because that contrast is the interesting signal.

In this paper, we introduce an efficient and general approach to learn the foreground-specific model without having to learn an accurate model of the background. Our approach is based on a method-of-moments that uses higher-order tensor decompositions for estimation [5]; we generalize the tensor decomposition technique to deal with our task of contrastive learning. Many other recent spectral learning algorithms for latent variable models are also based on the method-of-moments (*e.g.*, [6–13]), but their parameter estimation can not account for the asymmetry between foreground and background.

We demonstrate spectral contrastive learning through two concrete applications: contrastive topic modeling and contrastive genomics. In contrastive topic modeling we are given a foreground corpus of documents and a background corpus. We want to learn a fully generative topic model that explains the foreground-specific documents (the contrast). We show that even when the background is extremely sparse—too noisy to learn a good background topic model—our spectral contrast algorithm still recovers foreground-specific topics. In contrastive genomics, sequence data is modeled by HMMs. The foreground data is generated by a mixture of two HMMs; one is foreground-specific, and the other captures some background process. The background data is generated by this second HMM. Contrastive learning amplifies the foreground-specific signal, which have meaningful biological interpretations.

## 3 Contrastive topic modeling

To illustrate contrastive analysis and introduce tensor methods, we consider a simple topic model where each document is generated by exactly one topic. In LDA [1], this corresponds to setting the Dirichlet prior hyper-parameter $\alpha \to 0$. The techniques here can be extended to the general $\alpha > 0$ case using the moment transformations given in [10]. The generative topic model for a document is as follows.

- A word $x$ is represented by an indicator vector $e_x \in \mathbb{R}^D$ which is 1 in its $x$-th entry and 0 elsewhere. $D$ is the size of the vocabulary. A document is a bag-of-words and is represented by a vector $\mathsf{c} \in \mathbb{R}^D$ with non-negative integer word counts.

- A topic is first chosen according to the distribution on $[K] := \{1, 2, \ldots, K\}$ specified by the probability vector $w \in \mathbb{R}^K$.

- Given that the chosen topic is $t$, the words in the document are drawn independently from the distribution specified by the probability vector $\mu_t \in \mathbb{R}^D$.

Following previous work (*e.g.*, [10]) we assume that $\mu_1, \mu_2, \ldots, \mu_K$ are linearly independent probability vectors in $\mathbb{R}^D$. Let the foreground corpus of documents be generated by the mixture of $|A| + |B|$ topics $\{(\mu_t, w_t^{\mathsf{f}})\}_{t \in A} \cup \{(\mu_t, w_t^{\mathsf{f}})\}_{t \in B}$, and the background topics be generated by the mixture of $|B| + |C|$ topics $\{(\mu_t, w_t^{\mathsf{b}})\}_{t \in B} \cup \{(\mu_t, w_t^{\mathsf{b}})\}_{t \in C}$ (here, we assume $(A, B, C)$ is a non-trivial partition of $[K]$, and that $w_t^{\mathsf{f}}, w_t^{\mathsf{b}} > 0$ for all $t$). Our goal is to learn $\{(\mu_t, w_t^{\mathsf{f}})\}_{t \in A}$.

### 3.1 Moment decompositions

We use the symbol $\otimes$ to denote the tensor product of vectors, so $a \otimes b$ is the matrix whose $(i, j)$-th entry is $a_i b_j$, and $a \otimes b \otimes c$ is the third-order tensor whose $(i, j, k)$-th entry is $a_i b_j c_k$. Given a third-order tensor $T \in \mathbb{R}^{d_1 \times d_2 \times d_3}$ and vectors $a \in \mathbb{R}^{d_1}$, $b \in \mathbb{R}^{d_2}$, and $c \in \mathbb{R}^{d_3}$, we let $T(I, b, c) \in \mathbb{R}^{d_1}$ denote the vector whose $i$-th entry is $\sum_{j,k} T_{i,j,k} b_j c_k$, and $T(a, b, c)$ denote the scalar $\sum_{i,j,k} T_{i,j,k} a_i b_j c_k$.

We review the moments of the word observations in this model (see, *e.g.*, [10]). Let $x_1, x_2, x_3 \in [D]$ be three random words sampled from a random document generated by the foreground model (the discussion here also applies to the background model). The second-order (cross) moment matrix $M_2^{\mathsf{f}} := \mathbb{E}[e_{x_1} \otimes e_{x_2}]$ is the matrix whose $(i, j)$-th entry is the probability that $x_1 = i$ and $x_2 = j$. Similarly, the third-order (cross) moment tensor $M_3^{\mathsf{f}} := \mathbb{E}[e_{x_1} \otimes e_{x_2} \otimes e_{x_3}]$ is the

---

**Algorithm 1** Contrastive Topic Model estimator

---

**input** Foreground and background documents $\{\mathsf{c}_n^\mathsf{f}\}$, $\{\mathsf{c}_n^\mathsf{b}\}$; parameter $\gamma > 0$; number of topics $K$.
**output** Foreground-specific topics $\mathsf{Topics_f}$.
 1: Let $\hat{M}_2^\mathsf{f}$ and $\hat{M}_3^\mathsf{f}$ ($\hat{M}_2^\mathsf{b}$ and $\hat{M}_3^\mathsf{b}$) be the foreground (background) second- and third-order moment estimates based on $\{\mathsf{c}_n^\mathsf{f}\}$ ($\{\mathsf{c}_n^\mathsf{b}\}$), and let $\hat{M}_2 := \hat{M}_2^\mathsf{f} - \gamma\hat{M}_2^\mathsf{b}$ and $\hat{M}_3 := \hat{M}_3^\mathsf{f} - \gamma\hat{M}_3^\mathsf{b}$.
 2: Run Algorithm 2 with input $\hat{M}_2, \hat{M}_3, K$, and $N$ to obtain $\{(\hat{a}_t, \hat{\lambda}_t) : t \in [K]\}$.
 3: $\mathsf{Topics_f} := \{(\hat{a}_t/\|\hat{a}_t\|_1, 1/\hat{\lambda}_t^2) : t \in [K], \hat{\lambda}_t > 0\}$.

---

third-order tensor whose $(i, j, k)$-th entry is the probability that $x_1 = i, x_2 = j, x_3 = k$. Observe that for any $t \in A \cup B$, the $i$-th entry of $\mathbb{E}[e_{x_1}|\mathsf{topic} = t]$ is precisely the probability that $x_1 = i$ given $\mathsf{topic} = t$, which is $i$-th entry of $\mu_t$. Therefore, $\mathbb{E}[e_{x_1}|\mathsf{topic} = t] = \mu_t$. Since the words are independent given the topic, the $(i, j)$-th entry of $\mathbb{E}[e_{x_1} \otimes e_{x_2}|\mathsf{topic} = t]$ is the product of the $i$-th and $j$-th entry of $\mu_t$, *i.e.*, $\mathbb{E}[e_{x_1} \otimes e_{x_2}|\mathsf{topic} = t] = \mu_t \otimes \mu_t$. Similarly, $\mathbb{E}[e_{x_1} \otimes e_{x_2} \otimes e_{x_3}|\mathsf{topic} = t] = \mu_t \otimes \mu_t \otimes \mu_t$. Averaging over the choices of $t \in A \cup B$ with the weights $w_t^\mathsf{f}$ implies that the second- and third-order moments are

$$M_2^\mathsf{f} = \mathbb{E}[e_{x_1} \otimes e_{x_2}] = \sum_{t \in A \cup B} w_t^\mathsf{f} \, \mu_t \otimes \mu_t \quad \text{and} \quad M_3^\mathsf{f} = \mathbb{E}[e_{x_1} \otimes e_{x_2} \otimes e_{x_3}] = \sum_{t \in A \cup B} w_t^\mathsf{f} \, \mu_t \otimes \mu_t \otimes \mu_t.$$

(We discuss how to efficiently use documents of length $> 3$ in Section 5.2.) We can similarly decompose the background moments $M_2^b$ and $M_3^b$ in terms of tensors products of $\{\mu_t\}_{t \in B \cup C}$. These equations imply the following proposition (proved in Appendix A).

**Proposition 1.** *Let* $M_2^\mathsf{f}$, $M_3^\mathsf{f}$ *and* $M_2^\mathsf{b}$, $M_3^\mathsf{b}$ *be the second- and third-order moments from the foreground and background data, respectively. Define*

$$M_2 := M_2^\mathsf{f} - \gamma M_2^\mathsf{b} \quad \text{and} \quad M_3 := M_3^\mathsf{f} - \gamma M_3^\mathsf{b}.$$

*If* $\gamma \geq \max_{j \in B} w_j^\mathsf{f}/w_j^\mathsf{b}$, *then*

$$M_2 = \sum_{t=1}^{K} \omega_t \, \mu_t \otimes \mu_t \quad \text{and} \quad M_3 = \sum_{t=1}^{K} \omega_t \, \mu_t \otimes \mu_t \otimes \mu_t \tag{2}$$

*where* $\omega_t = w_t^\mathsf{f} > 0$ *for* $t \in A$ *(foreground-specific topic), and* $\omega_t \leq 0$ *for* $t \in B \cup C$.

**Using tensor decompositions.** Proposition 1 implies that the modified moments $M_2$ and $M_3$ have low-rank decompositions in which the components $t$ with positive multipliers $\omega_t$ correspond to the foreground-specific topics $\{(\mu_t, w_t^\mathsf{f})\}_{t \in A}$. A main technical innovation of this paper is a generalized tensor power method, described in Section 5, which takes as input (estimates of) second- and third-order tensors of the form in (2), and approximately recovers the individual components. We argue that under some natural conditions, the generalized power method is robust to large perturbations in $M_2^\mathsf{b}$ and $M_3^\mathsf{b}$, which suggests that foreground-specific topics can be learned even when it is not possible to accurately model the background. We use the generalized tensor power method to estimate the foreground-specific topics in our Contrastive Topic Model estimator (Algorithm 1). Proposition 1 gives the lower bound on $\gamma$; we empirically find that $\gamma \approx \max_{j \in B} w_j^\mathsf{f}/w_j^\mathsf{b}$ gives good results. When $\gamma$ is too large, the convergence of the tensor power worsens. Where possible in practice, we recommend using prior belief about foreground and background compositions to estimate $\max_{j \in B} w_j^\mathsf{f}/w_j^\mathsf{b}$, and then vary $\gamma$ as part of the exploratory analysis.

## 3.2 Experiments with contrastive topic modeling

We test our contrastive topic models on the RCV1 dataset, which consists of $\approx 800000$ news articles. Each document comes with multiple category labels (*e.g.*, economics, entertainment) and region labels (*e.g.*, USA, Europe, China). The corpus spans a large set of complex and overlapping categories, making this a good dataset to validate our contrastive learning algorithm.

In one set of experiments, we take documents associated with one region as the foreground corpus, and documents associated with a general theme, such as economics, as the background. The goal of the contrast is to find the region-specific topics which are not relevant to the background theme. The top half of Table 1 shows the example where we take USA-related documents as the foreground

| USA foreground | | | | | USA foreground, Economics background | | | | |
| --- | --- | --- | --- | --- | --- | --- | --- | --- | --- |
| percent | lbs | bond | million | stock | play | research | result | basketball | game |
| week | usda | municipal | week | price | round | science | hockey | game | run |
| rate | hog | index | sale | close | golf | cancer | nation | nation | hit |
| market | gilt | year | export | trade | open | cell | cap | la | win |
| wheat | barrow | trade | total | index | hole | study | ny | association | inn |
| China foreground | | | | | China foreground, Economics background | | | | |
| china | share | billion | shanghai | yuan | china | panda | earthquake | china | interest |
| ton | market | reserve | yuan | year | east | china | china | office | bond |
| percent | percent | bank | firm | bank | typhoon | year | office | court | million |
| import | million | balance | china | foreign | storm | xinhua | richt | smuggle | cost |
| alumin | trade | trade | exchange | invest | flood | zoo | scale | ship | moody |

Table 1: Top words from representative topics: foreground alone (left); foreground/background contrast (right). Each column corresponds to one topic.

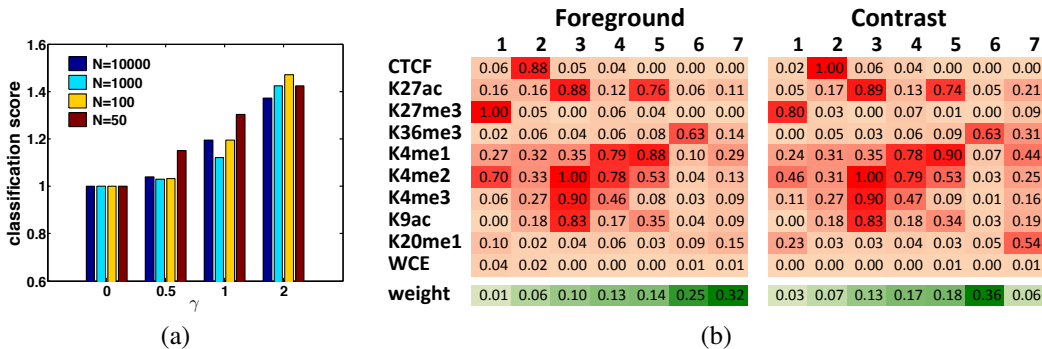

Figure 2: (a) Relative AUC as function of $\gamma$ (Sec. 3.2). (b) Emission probabilities of HMM states (Sec. 4).

and Economics as the background theme. We first set the contrast parameter $\gamma = 0$ in Algorithm 1; this learns the topics from the foreground dataset alone. Due to the composition of the corpus, the foreground topics for USA is dominated by topics relevant to stock markets and trade; representative topics and keywords are shown on the left of Table 1. Then we increase $\gamma$ to observe the effects of contrast. In the right half of Table 1, we show the heavily weighted topics and keywords for when $\gamma = 2$. The topics involving market and trade are also present in the background corpus, so their weights are reduced through contrast. Topics which are very USA-specific and distinct from economics rise to the top: basketball, baseball, scientific research, *etc*. A similar experiment with China-related articles as foreground, and the same economics themed background is shown in the bottom of Table 1.

These examples illustrate that Algorithm 1 learns topics which are unique to the foreground. To quantify this effect, we devised a specificity test. Using the RCV1 labels, we partition the foreground USA documents into two disjoint groups: documents with any economics-related labels (group 0) and the rest (group 1). Because Algorithm 1 learns the full probabilistic model, we use the inferred topic parameters to compute the marginal likelihood for each foreground document given the model. We then use the likelihood value to classify each foreground document as belonging to group 0 or 1. The performance of the classifier is summarized by the AUC score.

We first set $\gamma = 0$ and compute the AUC score, which corresponds to how well a topic model learned from only the foreground can distinguish between the two groups. We use this score as the baseline and normalize so it is equal to 1. The hope is that by using the background data, the contrastive model can better identify the documents that are generated by foreground-specific topics. Indeed, as $\gamma$ increases, the AUC score improves significantly over the benchmark (dark blue bars in Figure 2(a)). For $\gamma > 2$ we find that the foreground specific topics do not change qualitatively.

A major advantage of our approach is that we do not need to learn a very accurate background model to learn the contrast. To validate this, we down sample the background corpus to 1000, 100,

and 50 documents. This simulates settings where the background is very sparsely sampled, so it is not possible to learn a background model very accurately. Qualitatively, we observe that even with only 50 randomly sampled background documents, Algorithm 1 still recovers topics specific to USA and not related to Economics. At $\gamma = 2$, it learns sports and NASA/space as the most prominent foreground-specific topics. This is supported by the specificity test, where contrastive topic models with sparse background better identify foreground-specific documents relative to the $\gamma = 0$ (foreground data-only) model.

## 4  Contrastive Hidden Markov Models

Hidden Markov Models (HMMs) are commonly used to model sequence and time series data. For example, a biologist may collect several sequences from an experiment; some of the sequences are generated by a biological process of interest (modeled by an HMM), while others are generated by a different "background" process—*e.g.*, noise or a process that is not of primary interest.

Consider a simple generative process where foreground data are generated by a mixture of two HMMs: $(1 - \gamma)\,\mathrm{HMM}^A + \gamma\,\mathrm{HMM}^B$, and background data are generated by $\mathrm{HMM}^B$. The goal is to learn the parameters of $\mathrm{HMM}^A$, which models the biological process of interest. As we did for topic models, we can estimate a contrastive HMM by taking appropriate combinations of observable moments. Let $x_1^{\mathsf{f}}, x_2^{\mathsf{f}}, x_3^{\mathsf{f}}, \ldots$ be a random emission sequence in $\mathbb{R}^D$ generated by the foreground model $(1 - \gamma)\,\mathrm{HMM}^A + \gamma\,\mathrm{HMM}^B$, and $x_1^{\mathsf{b}}, x_2^{\mathsf{b}}, x_3^{\mathsf{b}}, \ldots$ be the sequence generated by the background model $\mathrm{HMM}^B$. Following [5], we estimate the following cross moment matrices and tensors: $M_{1,2}^{\mathsf{f}} := \mathbb{E}[x_1^{\mathsf{f}} \otimes x_2^{\mathsf{f}}]$, $M_{1,3}^{\mathsf{f}} := \mathbb{E}[x_1^{\mathsf{f}} \otimes x_3^{\mathsf{f}}]$, $M_{2,3}^{\mathsf{f}} := \mathbb{E}[x_2^{\mathsf{f}} \otimes x_3^{\mathsf{f}}]$, $M_{1,2,3}^{\mathsf{f}} := \mathbb{E}[x_1^{\mathsf{f}} \otimes x_2^{\mathsf{f}} \otimes x_3^{\mathsf{f}}]$, as well as the corresponding moments for the background model. This is similar to the estimation the word pair and triple frequencies in LDA. Here we only use the first three observations in the sequence, but it is also justifiable to average over all consecutive observation triplets [14]. Then, analogous to Proposition 1, we define the contrastive moments as $M_{u,v} := M_{u,v}^{\mathsf{f}} - \gamma M_{u,v}^{\mathsf{b}}$ (for $\{u, v\} \subset \{1, 2, 3\}$) and $M_{1,2,3} := M_{1,2,3}^{\mathsf{f}} - \gamma M_{1,2,3}^{\mathsf{b}}$. In the Appendix (Sec. D and Algorithm 3), we describe how to recover the foreground-specific model $\mathrm{HMM}^A$. The key technical difference from contrastive LDA lies in the asymmetric generalization of the Tensor Power Method of Algorithm 2.

**Application to contrastive genomics.**   For many biological problems, it is important to understand how signals in certain data are enriched relative to some related background data. For instance, we may want to contrast foreground data composed of gene expressions (or mutation rates, protein levels, *etc*) from one population against background data taken from (say) a control experiment, a different cell type, or a different time point. The contrastive analysis methods developed here can be a powerful exploratory tool for biology.

As a concrete illustration, we use spectral contrast to refine the characterization of chromatin states. The human genome consists of $\approx 3$ billion DNA bases, and has recently been shown that these bases can be naturally segmented into a handful of chromatin states [15, 16]. Each state describes a set of genomic properties: several states describe different active and regulatory features, while other states describe repressive features. The chromatin state varies across the genome, remaining constant for relatively short regions (say, several thousand bases). Learning the nature of the chromatin states is of great interest in genomics. The state-of-the-art approach for modeling chromatin states uses an HMM [16]. The observable data are, at every $200$ bases, a binary feature vector in $\{0, 1\}^{10}$. Each feature indicates the presence/absence of a specific chemical feature at that site (assumed independent given the chromatin state). This correspond to $\approx 15$ million observations across the genome, which are used to learn the parameters of an HMM. Each chromatin state corresponds to a latent state, characterized by a vector of 10 emission probabilities.

We take as foreground data the observations from exons, introns and promoters, which account for about $30\%$ of the genome; as background data, we take observations from intergenic regions. Because exons and introns are transcribed, we expect the foreground to be a mixture of functional chromatin states and spurious states due to noise, and expect more of the background observations to be due to non-functional process. The contrastive HMM should capture biologically meaningful signals in the foreground data. In Figure 2(b), we show the emission matrix for the foreground HMM and for the contrastive HMM. We learn $K = 7$ latent states, corresponding to 7 chromatin states.

---
**Algorithm 2** Generalized Tensor Power Method
---
**input** $\hat{M}_2 \in \mathbb{R}^{D \times D}$; $\hat{M}_3 \in \mathbb{R}^{D \times D \times D}$; target rank $K$; number of iterations $N$.
**output** Estimates $\{(\hat{a}_t, \hat{\lambda}_t) : t \in [K]\}$.

 1: Let $\hat{M}_2^\dagger :=$ Moore-Penrose pseudoinverse of rank $K$ approximation to $\hat{M}_2$; initialize $T := \hat{M}_3$.
 2: **for** $t = 1$ to $K$ **do**
 3:    Randomly draw $u^{(0)} \in \mathbb{R}^D$ from any distribution with full support in the range of $\hat{M}_2$.
 4:    Repeat power iteration update $N$ times: $u^{(i+1)} := T(I, \hat{M}_2^\dagger u^{(i)}, \hat{M}_2^\dagger u^{(i)})$.
 5:    $\hat{a}_t := u^{(N)}/|\langle u^{(N)}, \hat{M}_2^\dagger u^{(N)} \rangle|^{1/2}$; $\hat{\lambda}_t := T(\hat{M}_2^\dagger \hat{a}_t, \hat{M}_2^\dagger \hat{a}_t, \hat{M}_2^\dagger \hat{a}_t)$; $T := T - |\hat{\lambda}_t| \hat{a}_t \otimes \hat{a}_t \otimes \hat{a}_t$.
 6: **end for**
---

Each row is a chemical feature of the genome. The foreground states recover the known biological chromatin states from literature [16]. For example, state 6, with high emission for K36me3, is transcribed genes; state 5 is active enhancers; state 4 is poised enhancers. In the contrastive HMM, most of the states are the same as before. Interestingly, state 7, which is associated with feature K20me1, drops from the largest component of the foreground to a very small component of the contrast. This finding suggests that state 7 and K20me1 are less specific to gene bodies than previously thought [17], and raises more questions regarding its function, which is relatively unknown.

## 5   Generalized tensor power method

We now describe our general approach for tensor decomposition used in Algorithm 1. Let $a_1, a_2, \ldots, a_K \in \mathbb{R}^D$ be linearly independent vectors, and set $A := [a_1|a_2|\cdots|a_K]$. Let $M_2 := \sum_{i=1}^K \sigma_i a_i \otimes a_i$ and $M_3 := \sum_{i=1}^K \lambda_i a_i \otimes a_i \otimes a_i$, where $\sigma_i = \text{sign}(\lambda_i) \in \{\pm 1\}$. The goal is to recover $\{(a_t, \lambda_t) : t \in [K]\}$ from (estimates of) $M_2$ and $M_3$.

The following proposition shows that one of the vectors $a_i$ (and its associated $\lambda_i$) can be obtained from $M_2$ and $M_3$ using a simple power method similar to that from [5, 18] (note that which of the $K$ components is obtained depends on the initialization of the procedure). Note that the error $\varepsilon$ is exponentially small in $2^t$ after $t$ iterations, so the number of iterations required to converge is very small. Below, we use $(\cdot)^\dagger$ to denote the Moore-Penrose pseudoinverse.

**Proposition 2** (Informal statement). *Consider the sequence $u^{(0)}, u^{(1)}, \ldots$ in $\mathbb{R}^D$ determined by $u^{(i+1)} := M_3(I, M_2^\dagger u^{(i)}, M_2^\dagger u^{(i)})$ . Then for any $\varepsilon \in (0,1)$ and almost all $u^{(0)} \in \text{range}(A)$, there exists $t^* \in [K]$, $c_1, c_2 > 0$ (all depending on $u^{(0)}$ and $\{(\mu_t, \lambda_t) : t \in [K]\}$) such that $\|\tilde{u}^{(i)} - a_{t^*}\|^2 \le \varepsilon$ and $|\tilde{\lambda} - |\lambda_{t^*}|| \le |\lambda_{t^*}|\varepsilon + \max_{t \ne t^*} |\lambda_t|\varepsilon^{3/2}$ for $\varepsilon := c_1 \exp(-c_2 2^i)$, where $\tilde{u}^{(i)} := \sigma_{t^*} u^{(i)}/\|A^\dagger u^{(i)}\|$, and $\tilde{\lambda} := M_3(M_2^\dagger \tilde{u}^{(i)}, M_2^\dagger \tilde{u}^{(i)}, M_2^\dagger \tilde{u}^{(i)})$.*

See Appendix B for the formal statement and proof which give explicit dependencies. We use the iterations from Proposition 2 in our main decomposition algorithm (Algorithm 2), which is a variant of the main algorithm from [5]. The main difference is that we do not require $M_2$ to be positive semi-definite, which is essential for our application, but requires subtle modifications. For simplicity, we assume we run Algorithm 2 with exact moments $M_2$ and $M_3$ — a detailed perturbation analysis would be similar to that in [5] but is beyond the scope of this paper. Proposition 2 shows that a single component can be accurately recovered, and we use deflation to recover subsequent components (normalization and deflation is further discussed in Appendix B). As noted in [5], errors introduced in this deflation step have only a lower-order effect, and therefore it can be used reliably to recover all $K$ components. For increased robustness, we actually repeat steps 3–5 in Algorithm 2 several times, and use the results of the trial in which $|\hat{\lambda}_t|$ takes the median value.

### 5.1   Robustness to sparse background sampling

Algorithm 1 can recover the foreground-specific $\{\mu_t\}_{t \in A}$ even with relatively small numbers of background data. We can illustrate this robustness under the assumption that the support of the foreground-specific topics $S_0 := \cup_{t \in A} \text{supp}(\mu_t)$ is disjoint from that of the other topics $S_1 := \cup_{t \in B \cup C} \text{supp}(\mu_t)$ (similar to Brown clusters [19]). Suppose that $M_2^f$ is estimated accurately using a large sample of foreground documents. Then because $S_0$ and $S_1$ are disjoint, Algorithm 1

(using sufficiently large $\gamma$) will accurately recover the topics $\{(\mu_t, w_t^{\mathsf{f}}) : t \in A\}$ in $\mathsf{Topics_f}$. The remaining concern is that sampling errors will cause Algorithm 1 to mistakenly return additional topics in $\mathsf{Topics_f}$, namely the topics $t \in B \cup C$. It thus suffices to guarantee that the *signs* of the $\hat{\lambda}_t$ returned by Algorithm 2 are correct. The sample size requirement for this is *independent of the desired accuracy level for the foreground-specific topics*—it depends only on $\gamma$ and fixed properties of the background model.[1] As reported in Section 3.2, this robustness is borne out in our experiments.

## 5.2 Scalability

Our algorithms are scalable to large datasets when implemented to exploit sparsity and low-rank structure (each experiment we report runs on a standard laptop in a few minutes). Two important details are (i) how the moments $M_2$ and $M_3$ are represented, and (ii) how to execute the power iteration update in Algorithm 2. These issues are only briefly mentioned in [5] and without proof, so in this section, we address these issues in detail.

**Efficient moment estimates for topic models.** We first discuss how to represent empirical estimates of the second- and third-order moments $M_2^{\mathsf{f}}$ and $M_3^{\mathsf{f}}$ for the foreground documents (the same will hold for the background documents). Let document $n \in [N]$ have length $\ell_n$, and let $\mathsf{c}_n \in \mathbb{N}^D$ be its word count vector (its $i$-th entry $\mathsf{c}_n(i)$ is the number of times word $i$ appears in document $n$).

**Proposition 3** (Estimator for $M_2^{\mathsf{f}}$). *Assume $\ell_n \geq 2$. For any distinct $i, j \in [D]$, $\mathbb{E}[(\mathsf{c}_n(i)^2 - \mathsf{c}_n(i))/(\ell_n(\ell_n - 1))] = [M_2^{\mathsf{f}}]_{i,i}$ and $\mathbb{E}[\mathsf{c}_n(i)\mathsf{c}_n(j)/(\ell_n(\ell_n - 1))] = [M_2^{\mathsf{f}}]_{i,j}$.*

By Proposition 3, an unbiased estimator of $M_2^{\mathsf{f}}$ is $\hat{M}_2^{\mathsf{f}} := N^{-1} \sum_{n=1}^{N} (\ell_n(\ell_n - 1))^{-1} (\mathsf{c}_n \otimes \mathsf{c}_n - \mathrm{diag}(\mathsf{c}_n))$. Since $\hat{M}_2^{\mathsf{f}}$ is sum of sparse matrices, it can be represented efficiently, and we may use sparsity-aware methods for computing its low-rank spectral decompositions. It is similarly easy to obtain such a decomposition for $\hat{M}_2^{\mathsf{f}} - \gamma \hat{M}_2^{\mathsf{b}}$, from which one can compute its pseudoinverse and represent it in factored form as $PQ^{\top}$ for some $P, Q \in \mathbb{R}^{D \times K}$.

**Proposition 4** (Estimator for $M_3^{\mathsf{f}}$). *Assume $\ell_n \geq 3$. For any distinct $i, j, k \in [D]$, $\mathbb{E}[(\mathsf{c}_n(i)^3 - 3\mathsf{c}_n(i)^2 + 2\mathsf{c}_n(i))/(\ell_n(\ell_n - 1)(\ell_n - 2))] = [M_3^{\mathsf{f}}]_{i,i,i}$, $\mathbb{E}[(\mathsf{c}_n(i)^2\mathsf{c}_n(j) - \mathsf{c}_n(i)\mathsf{c}_n(j))/(\ell_n(\ell_n - 1)(\ell_n - 2))] = [M_3^{\mathsf{f}}]_{i,i,j}$, and $\mathbb{E}[(\mathsf{c}_n(i)\mathsf{c}_n(j)\mathsf{c}_n(k))/(\ell_n(\ell_n - 1)(\ell_n - 2))] = [M_3^{\mathsf{f}}]_{i,j,k}$.*

By Proposition 4, an unbiased estimator of $M_3^{\mathsf{f}}(I, v, v)$ for any vector $v \in \mathbb{R}^D$ is $\hat{M}_3^{\mathsf{f}}(I, v, v) := N^{-1} \sum_{n=1}^{N} (\ell_n(\ell_n - 1)(\ell_n - 2))^{-1} (\langle \mathsf{c}_n, v \rangle^2 \mathsf{c}_n - 2\langle \mathsf{c}_n, v \rangle(\mathsf{c}_n \circ v) - \langle \mathsf{c}_n, v \circ v \rangle \mathsf{c}_n + 2\mathsf{c}_n \circ v \circ v)$ (where $\circ$ denotes component-wise product of vectors). Let $\mathrm{nnz}(\mathsf{c}_n)$ be the number of non-zero entries in $\mathsf{c}_n$, then each term in the sum takes only $O(\mathrm{nnz}(\mathsf{c}_n))$ operations to compute. So the time to compute $\hat{M}_3^{\mathsf{f}}(I, v, v)$ is proportional to the number of non-zero entries of the term-document matrix, using just a single pass over the document corpus.

**Power iteration computation.** Each power iteration update in Algorithm 2 just requires the evaluating $\hat{M}_3^{\mathsf{f}}(I, v, v) - \gamma \hat{M}_3^{\mathsf{b}}(I, v, v)$ (one-pass linear time, as shown above) for $v := \hat{M}_2^{\dagger} u^{(i)}$, and computing the deflation $\sum_{\tau < t} \hat{\lambda}_\tau \langle \hat{a}_\tau, v \rangle^2 \hat{a}_\tau$ ($O(DK)$ time). Since $\hat{M}_2^{\dagger}$ is kept in rank-$K$ factored form, $v$ can also be computed in $O(DK)$ time.

## 6 Discussion

In this paper, we formalize a model of contrastive learning and introduce efficient spectral methods to learn the model parameters specific to the foreground. Experiments with contrastive topic modeling show that Algorithm 1 can learn foreground-specific topics even when the background data is noisy. Our application in contrastive genomics illustrates the utility of this method in exploratory analysis of biological data. The contrast identifies an intriguing change associated with K20me1, which can be followed up with biological experiments. While we have focused in this work on a natural contrast model for mixture models, we also discuss an alternative approach in Appendix E.

**Acknowledgement** This work was partially supported by DARPA Young Faculty Award DARPA N66001-12-1-4219.

## Footnotes

[1]For instance, if the background model consists only of one topic $\mu$, then the analyses from [5, 10] can be adapted to bound the sample size requirement by $O(1/\|\mu\|^6)$.

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
