[Supplementary Material · appendix_camera_ready.pdf]

# A  Proof of Proposition 1

*Proof of Proposition 1.* This follows from the observation that

$$M_2 = \sum_{t \in A} w_t^{\mathsf{f}} \, \mu_t \otimes \mu_t \; + \; \sum_{t \in B}(w_t^{\mathsf{f}} - \gamma w_t^{\mathsf{b}}) \, \mu_t \otimes \mu_t \; + \; \sum_{t \in C}(-\gamma w_t^{\mathsf{b}}) \, \mu_t \otimes \mu_t$$

$$M_3 = \sum_{t \in A} w_t^{\mathsf{f}} \, \mu_t \otimes \mu_t \otimes \mu_t \; + \; \sum_{t \in B}(w_t^{\mathsf{f}} - \gamma w_t^{\mathsf{b}}) \, \mu_t \otimes \mu_t \otimes \mu_t \; + \; \sum_{t \in C}(-\gamma w_t^{\mathsf{b}}) \, \mu_t \otimes \mu_t \otimes \mu_t$$

and

$$w_t^{\mathsf{f}} - \gamma w_t^{\mathsf{b}} \le 0 \; \forall t \in B \qquad \Longleftrightarrow \qquad \gamma \ge \max_{t \in B} w_t^{\mathsf{f}} / w_t^{\mathsf{b}}. \qquad \square$$

# B  Generalized tensor power method

**Normalization and deflation.** By Proposition 2, the first for-loop iteration of Algorithm 2 recovers $u^{(N)}$ very close to $\sigma_{i^*} a_{i^*}$ for some $i^* \in [K]$, up to positive scaling $s := \|A^\dagger u^{(N)}\|$. Because

$$1 = (\sigma_{i^*}\langle a_{i^*}, M_2^\dagger a_{i^*}\rangle)^{1/2} = |\langle a_{i^*}, M_2^\dagger a_{i^*}\rangle|^{1/2},$$

this scaling $s$ is close to $|\langle u^{(N)}, M_2^\dagger u^{(N)}\rangle|^{1/2}$, which is the normalization used in Algorithm 2. Thus, the estimates $\hat{a}_1$ and $\hat{\lambda}_1$ are close to $\sigma_{i^*} a_{i^*}$ and $\lambda_{i^*}$, respectively. For the next for-loop iteration, we want to execute the power iteration with a tensor close to $T - \lambda_{i^*} a_{i^*} \otimes a_{i^*} \otimes a_{i^*}$ in order to recover a component different from $a_{i^*}$. Therefore we use

$$M_3 - |\hat{\lambda}_1|\hat{a}_1 \otimes \hat{a}_1 \otimes \hat{a}_1 \approx M_3 - \sigma_{i^*}|\lambda_{i^*}|a_{i^*} \otimes a_{i^*} \otimes a_{i^*} = M_3 - \lambda_{i^*} a_{i^*} \otimes a_{i^*} \otimes a_{i^*}$$

(the crucial detail is the absolute value on $\hat{\lambda}_1$).

**Convergence analysis.**

**Proposition 5.** *Let $u^{(0)} \in \mathrm{range}(A)$, and consider the sequence determined by*

$$u^{(i+1)} := M_3(I, M_2^\dagger u^{(i)}, M_2^\dagger u^{(i)}).$$

*Define*

$$t^* := \arg\max_{t \in [K]} |\lambda_t \langle e_t, A^\dagger u^{(0)}\rangle|, \quad \rho := \max_{t \ne t^*}\left| \frac{\lambda_t \langle e_t, A^\dagger u^{(0)}\rangle}{\lambda_{t^*}\langle e_{t^*}, A^\dagger u^{(0)}\rangle}\right|, \quad \varepsilon := \rho^{2^{i+1}}\lambda_{t^*}^2 \sum_{t \ne t^*} \lambda_t^{-2},$$

$$\tilde{u}^{(i)} := \sigma_{t^*} u^{(i)}/\|A^\dagger u^{(i)}\|.$$

*Then*

$$\|A^\dagger(\tilde{u}^{(i)} - a_{t^*})\|^2 \le 2\varepsilon,$$

$$\left| M_3(M_2^\dagger \tilde{u}^{(i)}, M_2^\dagger \tilde{u}^{(i)}, M_2^\dagger \tilde{u}^{(i)}) - |\lambda_{t^*}| \right| \le |\lambda_{t^*}| \cdot \varepsilon + \max_{t \ne t^*}|\lambda_t| \cdot \varepsilon^{1.5}.$$

*Proof.* Define $f_t := \langle e_t, A^\dagger u^{(0)}\rangle$, and without loss of generality, assume $|\lambda_1 f_1| \ge |\lambda_2 f_2| \ge \cdots \ge |\lambda_K f_K|$. Then, using the definition $u^{(1)} = M_3(I, M_2^\dagger u^{(0)}, M_2^\dagger u^{(0)})$ and the facts that $A$ has full column rank and $\Sigma$ is invertible, we have

$$u^{(1)} = \sum_{t=1}^K \lambda_t \langle a_t, M_2^\dagger u^{(0)}\rangle^2 a_t$$

$$= \sum_{t=1}^K \lambda_t \langle a_t, (A^\top)^\dagger \Sigma^{-1} A^\dagger u^{(0)}\rangle^2 a_t$$

$$= \sum_{t=1}^K \lambda_t \sigma_t^{-2} \langle e_t, A^\dagger u^{(0)}\rangle^2 a_t$$

$$= \sum_{t=1}^K \lambda_t f_t^2 a_t,$$

which implies $\langle e_t, A^\dagger u^{(0)} \rangle = \lambda_t f_t^2$. By induction, $\langle e_t, A^\dagger u^{(i)} \rangle = \lambda_t^{2^i-1} f_t^{2^i}$. Therefore

$$1 - \langle e_1, A^\dagger \tilde{u}^{(i)} \rangle^2 = 1 - \frac{\langle e_1, A^\dagger u^{(i)} \rangle^2}{\sum_{t=1}^K \langle e_t, A^\dagger u^{(i)} \rangle^2} = 1 - \frac{|\lambda_1|^{2^{i+1}-2} f_1^{2^{i+1}}}{\sum_{t=1}^K |\lambda_t|^{2^{i+1}-2} f_t^{2^{i+1}}} \le \rho^{2^{i+1}} \lambda_1^2 \sum_{t=2}^K \lambda_t^{-2} = \varepsilon.$$

Moreover, $\langle e_1, A^\dagger \tilde{u}^{(i)} \rangle = |\lambda_1|^{2^i-1} f_1^{2^i} / \sqrt{\sum_{t=1}^K |\lambda_t|^{2^{i+1}-2} f_t^{2^{i+1}}} \in [0,1]$, so $\langle e_1, A^\dagger \tilde{u}^{(i)} \rangle \ge \langle e_1, A^\dagger \tilde{u}^{(i)} \rangle^2$. Therefore, using the fact that $\|A^\dagger \tilde{u}^{(i)}\| = \|A^\dagger a_1\| = 1$, we can bound $\|A^\dagger (\tilde{u}^{(i)} - a_1)\|^2$ as

$$\|A^\dagger (\tilde{u}^{(i)} - a_1)\|^2 = 2(1 - \langle a_1, (AA^\top)^\dagger \tilde{u}^{(i)} \rangle) = 2(1 - \langle e_1, A^\dagger \tilde{u}^{(i)} \rangle) \le 2(1 - \langle e_1, A^\dagger \tilde{u}^{(i)} \rangle^2) \le 2\varepsilon.$$

It remains to show that $M_3(M_2^\dagger \tilde{u}^{(i)}, M_2^\dagger \tilde{u}^{(i)}, M_2^\dagger \tilde{u}^{(i)})$ is close to $|\lambda_1|$. We have that

$$\begin{aligned}
M_3(M_2^\dagger \tilde{u}^{(i)}, M_2^\dagger \tilde{u}^{(i)}, M_2^\dagger \tilde{u}^{(i)}) &= \sum_{t=1}^K \lambda_t \langle a_t, M_2^\dagger \tilde{u}^{(i)} \rangle^3 \\
&= \sum_{t=1}^K |\lambda_t| \langle e_t, A^\dagger \tilde{u}^{(i)} \rangle^3 \\
&= |\lambda_1| \langle e_1, A^\dagger \tilde{u}^{(i)} \rangle + \sigma_1 \sum_{t=2}^K |\lambda_t| \left( \frac{\langle e_t, A^\dagger u^{(i)} \rangle}{\sqrt{\sum_{j=1}^K \langle e_j, A^\dagger u^{(i)} \rangle^2}} \right)^3.
\end{aligned}$$

Since $(1 + \langle e_1, A^\dagger \tilde{u}^{(i)} \rangle)(1 - \langle e_1, A^\dagger \tilde{u}^{(i)} \rangle) = 1 - \langle e_1, A^\dagger \tilde{u}^{(i)} \rangle^2 \le \varepsilon$ and $\langle e_1, A^\dagger \tilde{u}^{(i)} \rangle \in [0,1]$, it follows that $|1 - \langle e_1, A^\dagger \tilde{u}^{(i)} \rangle| = (1 - \langle e_1, A^\dagger \tilde{u}^{(i)} \rangle) \le \varepsilon/(1 + \langle e_1, A^\dagger \tilde{u}^{(i)} \rangle) \le \varepsilon$. Furthermore, by Hölder's inequality, the triangle inequality, and the fact that $(\sum_t |v_t|^3)^{1/3} \le (\sum_t v_t^2)^{1/2}$,

$$\begin{aligned}
\left| \sum_{t=2}^K |\lambda_t| \left( \frac{\langle e_t, A^\dagger u^{(i)} \rangle}{\sqrt{\sum_{j=1}^K \langle e_j, A^\dagger u^{(i)} \rangle^2}} \right)^3 \right| &\le \left( \max_{t>1} |\lambda_t| \right) \frac{\sum_{t=2}^K |\langle e_t, A^\dagger u^{(i)} \rangle|^3}{\left( \sum_{j=1}^K \langle e_j, A^\dagger u^{(i)} \rangle^2 \right)^{3/2}} \\
&\le \max_{t>1} |\lambda_t| \left( \frac{\sum_{t=2}^K \langle e_t, A^\dagger u^{(i)} \rangle^2}{\sum_{j=1}^K \langle e_j, A^\dagger u^{(i)} \rangle^2} \right)^{3/2} \\
&\le \max_{t>1} |\lambda_t| \varepsilon^{3/2}.
\end{aligned}$$

Thus, again by the triangle inequality,

$$\left| M_3(M_2^\dagger \tilde{u}^{(i)}, M_2^\dagger \tilde{u}^{(i)}, M_2^\dagger \tilde{u}^{(i)}) - |\lambda_1| \right| \le |\lambda_1| \varepsilon + \max_{t>1} |\lambda_t| \varepsilon^{3/2}. \qquad \square$$

## C  Moment estimators

In the proofs of Propositions 3 and 4, we let $x_{n,1}, x_{n,2}, \ldots, x_{n,\ell_n} \in [D]$ be the words in document $n$, so $\mathsf{c}_n := \sum_{i=1}^{\ell_n} e_{x_{n,i}}$.

*Proof of Proposition 3.* For any $i \in [D]$,

$$\begin{aligned}
\mathbb{E}\left[ \mathsf{c}_n(i)^2 - \mathsf{c}_n(i) \right] &= \mathbb{E}\left[ \left( \sum_{p=1}^{\ell_n} x_{n,p}(i) \right)^2 - \sum_{p=1}^{\ell_n} x_{n,p}(i) \right] \\
&= \mathbb{E}\left[ \sum_{p=1}^{\ell_n} x_{n,p}(i)^2 + 2\sum_{p<q} x_{n,p}(i) x_{n,q}(i) - \sum_{p=1}^{\ell_n} x_{n,p}(i) \right] \\
&= 2\sum_{p<q} \mathbb{E}\left[ x_{n,p}(i) x_{n,q}(i) \right] \quad \text{(since } x_{n,p}(i)^2 = x_{n,p}(i)) \\
&= \ell_n(\ell_n - 1)[M_2^f]_{i,i}.
\end{aligned}$$

For $i \neq j$,

$$
\begin{aligned}
\mathbb{E}\Big[\mathsf{c}_n(i)\mathsf{c}_n(j)\Big] &= \mathbb{E}\left[\sum_{p=1}^{\ell_n} x_{n,p}(i) \sum_{q=1}^{\ell_n} x_{n,q}(j)\right] \\
&= \mathbb{E}\left[\sum_{p=1}^{\ell_n} x_{n,p}(i)x_{n,p}(j) + \sum_{p \neq q} x_{n,p}(i)x_{n,q}(j)\right] \\
&= \sum_{p \neq q} \mathbb{E}\Big[x_{n,p}(i)x_{n,q}(j)\Big] \quad \text{(since } x_{n,p}(i)x_{n,p}(j) = 0 \text{ for } i \neq j) \\
&= \ell_n(\ell_n - 1)[M_2^{\mathsf{f}}]_{i,j}. \hspace{5cm} \square
\end{aligned}
$$

*Proof of Proposition 4.* For any $i \in [D]$,

$$
\begin{aligned}
\mathbb{E}\Big[\mathsf{c}_n(i)^3 - 3\mathsf{c}_n(i)^2 + 2\mathsf{c}_n(i)\Big] &= \mathbb{E}\left[\left(\sum_{p=1}^{\ell_n} x_{n,p}(i)\right)^3 - 3\left(\sum_{p=1}^{\ell_n} x_{n,p}(i)\right)^2 + 2\left(\sum_{p=1}^{\ell_n} x_{n,p}(i)\right)\right] \\
&= \mathbb{E}\left[\sum_{p=1}^{\ell_n} x_{n,p}(i)^3 + 3\sum_{p<q}\Big(x_{n,p}(i)^2 x_{n,q}(i) + x_{n,p}(i)x_{n,q}(i)^2\Big)\right. \\
&\qquad + 6\sum_{p<q<r} x_{n,p}(i)x_{n,q}(i)x_{n,r}(i) \\
&\qquad \left. - 3\sum_{p=1}^{\ell_n} x_{n,p}(i)^2 - 6\sum_{p<q} x_{n,p}(i)x_{n,q}(i) + 2\sum_{p=1}^{\ell_n} x_{n,p}(i)\right] \\
&= 6\sum_{p<q<r} \mathbb{E}\Big[x_{n,p}(i)x_{n,q}(i)x_{n,r}(i)\Big] \quad \text{(since } x_{n,p}(i)^3 = x_{n,p}(i)^2 = x_{n,p}(i)) \\
&= \ell_n(\ell_n-1)(\ell_n-2)[M_3^{\mathsf{f}}]_{i,i,i}.
\end{aligned}
$$

For $i \neq j$,

$$
\begin{aligned}
\mathbb{E}\Big[\mathsf{c}_n(i)^2\mathsf{c}_n(j) - \mathsf{c}_n(i)\mathsf{c}_n(j)\Big] &= \mathbb{E}\left[\left(\sum_{p=1}^{\ell_n} x_{n,p}(i)\right)^2\left(\sum_{r=1}^{\ell_n} x_{n,r}(j)\right) - \left(\sum_{p=1}^{\ell_n} x_{n,p}(i)\right)\left(\sum_{q=1}^{\ell_n} x_{n,q}(j)\right)\right] \\
&= \mathbb{E}\left[\sum_{p=1}^{\ell_n} x_p(i)^2 x_p(j) + \sum_{p \neq q} x_p(i)^2 x_q(j)\right. \\
&\qquad + \sum_{p \neq q}\Big(x_p(i)x_p(j)x_q(i) + x_p(i)x_q(i)x_q(j)\Big) + \sum_{p \neq q \neq r} x_p(i)x_q(i)x_r(j) \\
&\qquad \left. - \sum_{p=1}^{\ell_n} x_p(i)x_p(j) - \sum_{p \neq q} x_p(i)x_q(j)\right] \\
&= \mathbb{E}\left[\sum_{p \neq q} x_p(i)^2 x_q(j) + \sum_{p \neq q \neq r} x_p(i)x_q(i)x_r(j) - \sum_{p \neq q} x_p(i)x_q(j)\right] \\
&= \sum_{p \neq q \neq r} \mathbb{E}\Big[x_p(i)x_q(i)x_r(j)\Big] \\
&= \ell_n(\ell_n-1)(\ell_n-2)[M_3^{\mathsf{f}}]_{i,i,j},
\end{aligned}
$$

---
**Algorithm 3** Asymmetric Generalized Tensor Power Method
---
**input** $\hat{M}_{a,b} \in \mathbb{R}^{D_a \times D_b}$; $\hat{M}_{a,c} \in \mathbb{R}^{D_a \times D_c}$; $\hat{M}_{b,c} \in \mathbb{R}^{D_b \times D_c}$; $\hat{M}_{a,b,c} \in \mathbb{R}^{D_a \times D_b \times D_c}$; target rank $K$; number of iterations $N$.

**output** Estimates $\{(\hat{a}_t, \hat{b}_t, \hat{c}_t, \hat{\lambda}_t) : t \in [K]\}$.

1: Let $\hat{M}_{a,b}^\dagger :=$ Moore-Penrose pseudoinverse of rank $K$ approximation to $\hat{M}_{a,b}$; similarly define $\hat{M}_{a,c}^\dagger$ and $\hat{M}_{b,c}^\dagger$; let $\hat{M}_{a,a}^\dagger := \hat{M}_{b,a}^\dagger \hat{M}_{b,c} \hat{M}_{a,c}^\dagger$; initialize $T := \hat{M}_{a,b,c}$.

2: **for** $t = 1$ to $K$ **do**

3:     Randomly draw $u^{(0)} \in \mathbb{R}^D$ from any distribution with full support in the range of $\hat{M}_{a,b}$.

4:     Repeat power iteration update $N$ times: $u^{(i+1)} := T(I, \hat{M}_{a,b}^\dagger u^{(i)}, \hat{M}_{a,c}^\dagger u^{(i)})$.

5:     $\hat{a}_t := u^{(N)}/|\langle u^{(N)}, \hat{M}_{a,a}^\dagger u^{(N)}\rangle|^{1/2}$; $\hat{b}_t := \hat{M}_{b,c}\hat{M}_{a,c}^\dagger\hat{a}_t$; $\hat{c}_t := \hat{M}_{c,b}\hat{M}_{a,b}^\dagger\hat{a}_t$; $\hat{\lambda}_t :=$ $T(\hat{M}_{a,a}^\dagger\hat{a}_t, \hat{M}_{a,b}^\dagger\hat{a}_t, \hat{M}_{a,c}^\dagger\hat{a}_t)$; $T := T - |\hat{\lambda}_t|\hat{a}_t \otimes \hat{b}_t \otimes \hat{c}_t$.

6: **end for**
---

where the third step uses the fact that $x_{n,p}(i)x_{n,q}(j) = 0$ for $i \neq j$, and the fourth step uses the fact that $x_{n,p}(i)^2 = x_{n,p}(i)$. Finally, for $i \neq j \neq k$,

$$\mathbb{E}\Big[c_n(i)c_n(j)c_n(k)\Big] = \mathbb{E}\left[\left(\sum_{p=1}^{\ell_n} x_{n,p}(i)\right)\left(\sum_{q=1}^{\ell_n} x_{n,q}(j)\right)\left(\sum_{r=1}^{\ell_n} x_{n,r}(k)\right)\right]$$

$$= \sum_{p \neq q \neq r} \mathbb{E}\Big[x_{n,p}(i)x_{n,q}(j)x_{n,r}(k)\Big]$$

$$= \ell_n(\ell_n - 1)(\ell_n - 2)[M_3^f]_{i,j,k},$$

where we use the same two facts in the second step. $\qquad\square$

## D    Asymmetric generalized tensor power method

Let $\{a_1, a_2, \ldots, a_K\} \subset \mathbb{R}^{D_a}$, $\{b_1, b_2, \ldots, b_K\} \subset \mathbb{R}^{D_b}$, and $\{c_1, c_2, \ldots, c_K\} \subset \mathbb{R}^{D_c}$ be sets of linearly independent vectors. Let $M_{u,v} := \sum_{t=1}^K \sigma_t u_t v_t^\top$ for $(u,v) \in \{a,b,c\} \times \{a,b,c\}$, and $M_{a,b,c} := \sum_{t=1}^K \lambda_t a_t \otimes b_t \otimes c_t$, where $\sigma_t = \text{sign}(\lambda_t) \in \{\pm 1\}$. Given (estimates of) $M_{a,b}, M_{a,c}, M_{b,c}, M_{a,b,c}$, Algorithm 3 approximates $\{(a_t, b_t, c_t, \lambda_t) : t \in [K]\}$. The proof of convergence (assuming exact estimates of $M_{a,b}, M_{a,c}, M_{b,c}, M_{a,b,c}$) is very similar to Proposition 2 and is thus omitted.

## E    An alternative model of contrast

We consider a different generative model in which the topic of a document is a simple (fixed) mixture of a foreground topic and a background topic (say, $0.9\mu_t^f + 0.1\mu_{t'}^b$ with probability $w_{t,t'}$, for $t \in [K^f]$ and $t \in [K^b]$). One can treat this using the previous model with $K = K^f K^b$ topics, but there are really only $K^f + K^b$ topics. Using auxiliary background data which is modeled by a topic model over just the background topics $\{\mu_{t'}^b : t' \in [K^b]\}$, it is possible to determine an orthogonal projector $\Pi \in \mathbb{R}^{D \times D}$ for the range of the second-order moments, which approximately captures the span of the $\{\mu_{t'}^b\}$. Then, the projection $I - \Pi$ can be applied to the second- and third-order moments of the foreground documents (which is generated by mixed topics) to annihilate the background topic contributions: $(I - \Pi)(0.9\mu_t^f + 0.1\mu_{t'}^b) = 0.9(I - \Pi)\mu_t^f$. If, in addition, the support of the foreground topics and background topics are disjoint (as in Brown clusters), then $(I - \Pi)\mu_t^f = \mu_t^f$. Therefore, one can directly estimate the $K^f$ foreground topics using the foreground data. Moreover, we do not need to fully estimate the model for the background documents, as we only need the second-order (but not third-order) moments to determine $\Pi$.

We used this model to conduct experiments similar to those reported in Section 3.2 on the RCV1 dataset, and observed qualitatively similar results, but it was less numerically stable compared to Algorithm 1. Developing better estimators for this model is a promising direction of future research.