[Reviews · NeurIPS 2013]

Submitted by Assigned_Reviewer_4

This paper develops a model for "contrastive learning" which aims to capture structure in a "foreground" set of target observations as opposed to generic "background" structure. The moments of the foreground and background are expressed as tensor products, estimation is then based on the difference between the moments of the foreground and moments of the background.

The concept of contrastive learning appears to be novel and of general interest, at least conceptually. The authors compare the idea conceptually to PCA and linear discriminant analysis, I would be interested in seeing a comparison of performance as well. Does contrastive learning actually pick up different structure, and when?

This is a clear and well written paper within the space constraints. However, with that said, this paper suffers from having too much stuffed into 8 pages. My main complaint is simply that I want more detail.

The section describing contrastive hidden Markov models, and the application to genomics does not have enough detail. The equations in the paragraph lines 286-300, really don't have enough detail to figure out what's different and what's the same as in the topic model application. The results in lines 336-342 also lack detail and context. And it's very confusing for readers that Figure 2a goes with the topic modeling application (at least according to line 269), but Figure 2b goes with the HMM application. Even worse, you don't mention this in the caption!!

line 295: the word "justifiable" is ambiguous. Does that mean that's what you did? Or does that mean, you could if you wanted to? But again, this whole paragraph needs more detail.

The "specificity" test described on lines 258-269 seems slightly bizarre to me. Though, I admit that this may be due again to a lack of detail under the space constraints. Why is it informative that a model trained only on foreground data has poor performance when used as a classifier for foreground/background? Plus the purpose of this method is not to classify foreground v. background, it is to analyze the variation in the foreground (lines 050-052, lines 073-075, or even line 106).

I would find a simulation study that demonstrates the method's ability to pick up patterns in foreground data, much more convincing than this specificity test.


Small item:
line 376: "…under the [assumption] that the support…"
Summary: The idea of separating foreground structure from background structure appears to be novel, and of general interest. However, this paper lacks detail because it is clearly up against the strict space constraints.

Submitted by Assigned_Reviewer_5

Summary:
--------

The authors introduce the paradigm of "contrastive learning" for mixture
models, which aims at learning the parameters of mixture components that are
unique to "foreground" data and not present in "background" data. They
present a spectral approach to contrastive learning to mixtures over models which
have a spectral learning algorithm. They illustrate their approach with examples
of contrastive learning of topic models and HMMs.



Comments:
---------

The main idea of the paper, the "contrastive learning" framework, is well
motivated and might indeed be of practical use (although it looks a bit tailored
to the "spectral trick" of subtracting the moment matrices/tensors). Both
numerical experiments (LDA, HMM) nicely illustrate that contrastive learning
might be an interesting and worthwhile approach to pursue. The proposed
algorithm builds on very recent developments in spectral learning, which makes
it of interest to a broader NIPS audience. It seems to be technically sound,
although I lack the expertise to be able to check all the technical
contributions in detail. The paper is clearly structured and well written.


My comments mainly focus on the fact that the authors could have made a better
effort to put their contribution into the bigger machine learning context:

1) The authors state (p2, last line and p3) that the direct, possibly naive way
of doing contrastive learning would be to learn two models, one
for the background and one for the foreground, and then to isolate the foreground
specific components. They state advantages of their approach, especially
robustness, however they never actually show these experimentally in
comparison to the naive approach. The authors should compare their algorithm on
their two numerical experiments with a "naive" contrastive model: A Mixture
model each for the for- and the background with (partially) shared parameters
learned by EM (or a similar model). Is the spectral method better / faster
/ more robust?

2) pp7, section 5.1: It is conceivable that a "naive contrastive model" (see
above) could also be robust to sparse sampling of the background model under
the assumption of disjoint support of background and for ground-specific
components. It is important to also demonstrate the robustness of the proposed
algorithm on data relative to the naive model.

3) Learning the statistics of some signal (foreground) that is embedded in some
noise (background) is of course a major theme in signal processing / stats /
machine learning. It would have been nice if the authors briefly compared their
approach to other methods in the introduction or discussion.
Summary: The paper contains intriguing ideas, is well written and of interest to the NIPS
audience. It would have benefitted from more thorough experiments.

Submitted by Assigned_Reviewer_7

[Summary]
This paper proposes a method for what the authors call contrastive learning
on the basis of a power-iteration based tensor decomposition method for
latent variable models. An experimental result on a simple topic model
is shown to demonstrate usefulness of the proposal. Efficiency of the
proposal in gemonic data analysis is suggested via an experiment on chromatin
states.

[Quality]
The proposed method is interesting from application point of view, in
particular its demonstrated efficiency in finding topics specific to
a certain foreground corpus in contrast to background. From the theoretical
viewpoint, however, the argument in Section 5 is on something which seems
different from what has been discussed up to Section 4, in particular
in the definition of M_2 given in the line numbered 348-349, where the
coefficients are \sigma_i=sign(\lambda_i) rather than \lambda_i themselves.
This incoherence in the descriptions makes possible contributions of this
paper somewhat obscure.

[Clarity]
In Figure 1, I do not understand what the red and blue lines represent.
The example is not the same as the model considered in this paper (mixture
models), which I feel makes it a rather bad example for explanation
purposes.
A document is defined as a bag-of-words, but in Sect. 3.1 "the first,
second, and third words" in a document are used in the description,
which is confusing.
In the line numbered 072 "learn amplify".
In the first line of page 3, "just from {x_n^f}," the superscript should
read b instead of f.
In the line numbered 276, "under the that ...".
In the line numbered 317, "Each chromatin state correspond(s) to".
In the line numbered 390, "to compute execute".
After Proposition 4, \circ and "nnz" are used without explicit definitions.

[Originality]
The idea of applying a power-method based tensor decomposition method
to the contrastive learning is considered original.

[Significance]
This paper is of importance in application point of view, in that
the proposed approach to contrastive learning has successfully demonstrated
its efficiency even in an application of a simple topic model. It is rather
difficult to judge the significance of this paper in the theoretical side,
mainly because of the incoherence mentioned above, as well as a rather brief
explanation given in the main paper.
Summary: This paper proposes an intersting approach to contrastive
learning using a power-method based tensor decomposition method. Demonstrated
usefulness of the proposal via an experiment on a simple topic model is
interesting.

Submitted by Assigned_Reviewer_8

This paper tackles the problem of learning a mixture model to capture
properties of "foreground data" which are not present in background
data. The proposed technique extends recent work on method of moments
and tensor factorization. The novelty in the paper is (i) the
observation that one can subtract moments before factorization and (ii)
developing a variant of the robust power method from Anandkumar et al. 2013
that doesn't require positive semidefiniteness.

The idea presented in the paper is simple, natural, and novel. The
main weakness is that the paper doesn't really compare the proposed
method with several natural baselines either theoretically or
empirically.

073: the paper says that the stated goal is not to discriminate between
foreground and background, but isn't this exactly the intuition of
being contrastive?

383: In the case where the foreground words are disjoint from the rest
(B and C), the intuition that you only need enough background data to
get the signs of the topics in B and C correct seems right. However,
the paragraph then talks about the accuracy of the foreground topics,
which doesn't relevant. I would have expected to see some statement
characterizing the number of samples you need to get the sign correct.

Also, I normally think of the foreground as being captured by a modest
number of topics and that the background is enormous and requires a lot
more, so $K$ would have to be pretty large, close to $D$. So footnote
1, which talks about one topic, seems pretty unmotivated.

The fact that power iteration requires multiple passes over the
training data ($K$ times the number of power iteration updates times
the number of restarts of the power tensor method) is somewhat
discouraging, since one of the benefits of these method of moments
approaches is that you can take one pass over the data. Could you
randomly project the data onto $O(K)$ dimensions, building up a much
smaller dense tensor that you can store in memory?

Baseline 1: estimate the foreground model and background models
separately. Under the assumption that the model is well-specified,
then the normal spectral method of moments will provide a consistent
estimator of both models, from which it is easy to detect which are the
common topics. What can you say about this method theoretically (it is
consistent after all) and empirically?

Baseline 2: simply use EM to do parameter estimation in the full
generative model. Is local optima is actually a problem in this
setting? My experience is that EM works pretty well anyway, and any
sort of empirical statement about this approach must compare with EM.
Even if EM outperforms spectral, you can always use the latter as
initialization to the former.

Theoretically, I think it's interesting to think about the behavior of
the maximum likelihood estimator in this case, which in some sense
tries to spread its modeling efforts evenly, whereas the contrastive
estimator seems more targeted towards what you want. So there might be
a regime that the contrastive estimator is better, even theoretically?

One piece of work which is not directly relevant but has a similar
high-level flavor is contrastive estimation:
http://www.cs.cmu.edu/~nasmith/papers/smith+eisner.acl05.pdf
Actually, now that I think about it, the intuition is somewhat
different - instead of using all the background data, we are using a
subset which is "close" to the foreground data.
Summary: This paper provides a new spectral method of moments estimator for
estimating a mixture model on foreground data, contrasting against
background data. Though there could more comparison to baselines,
overall the idea put forth is novel and interesting.
Author Feedback

Author rebuttal: We thank all the reviewers for their thoughtful comments and suggestions. As the reviewers mention, we had to leave out some discussions due to space limitation. We'll incorporate more descriptions and other suggestions in the revision.

Reviewer4
PCA or linear discriminant analysis do not yield parameters which can be interpreted as a topic's distribution over words or the emission matrix in HMM. This makes it difficult to compare our method against them.

We agree that the HMM section needs more description and will include them in revision. Will fix the caption of Figure 2. On Line 295, we say that the reference [13] provides justification (under a "fast mixing" condition) for averaging over all consecutive observation triplets.

Classification accuracy is only a surrogate for measuring how well a model captures topics specific to the foreground. Our experiments show that the contrastive model assigns higher probability to documents that are about USA but not economics and lower probability to documents that are about both "USA and economics" (when "economics" is the background). The model trained just on foreground is a baseline.

Unfortunately we did not have room to include a simulation study (although we did conduct one), but will try to include it in a revision.

Reviewer5
Good suggestion on placing this work in the wider context, we'll include more discussions in revision. Some related techniques in stats are anomaly and novelty detection. Key differences are that typically in those settings the background data is abundant and they don't directly learn a generative model of the anomaly.

Constructing a generative model that accounts for both foreground and background is an interesting research direction. A key advantage of our approach is that we do not need to learn a good model of the background. We have experiments showing that this is more efficient when the background is large and complex, since we only need to compute its moments. When the background is very sparse and we apply LDA, we find that it doesn't capture coherent topics but just sets of keywords. It's not clear how to use this to filter out shared topics in the foreground in a principled way. Will include more discussion of this.

Reviewer_7
Thanks for your review and interest in the applications. The key theoretical contributions are that we developed fast new tensor algorithm to deal with indefinite tensors, and proved how this allows us to recover a foreground specific model. This extends existing tensor methods which require positive semidefiniteness. We had to put most of the theoretical discussion in the supplement due to space; will try to clarify this in revision.

The model in Sec 3,4 is consistent with Sec 5. To tranform Eqn.2 to line 349, we map a_i to mu_i*sqrt(|w_i|), sigma_i to sign(w_i) and lambda_i to sign(w_i)/sqrt(|w_i|). We used this new notation in Sec 5 to convey that its results are generally applicable to indefinite tensors. Prop2 shows how we can recover {a_t, lambda_t}, and from this we can reconstruct {w_t, mu_t} for LDA. We will clarify the notations in revision.

Contrastive HMM and LDA are special cases of indefinite tensors. The key idea is that when we take contrast of moments, M_2^f -gamma M_2^b and M_3^f - gamma M_3^b, the foreground specific components have positive sign and the background have negative sign.

In the bag-of-words representation, we can arbitrarily order the words and call one word the first, another word the second, and so on. This is used to explain the structure of the moments (similar to [9]). Sec 5.2 shows how to work directly with the word count vector to avoid explicitly enumerating triples of words in the document.

Fig 1 is meant to be a toy illustration to motivate contrast. The idea is that we would like to learn a subspace (represented by the blue line in (c)) that captures variance of the foreground data that is not already captured by the background projection (represented by the red line). We'll clarify this.

Reviewer_8
Our motivation for contrastive learning is to learn a full generative model that captures the data structures specific to the foreground (FG), rather than to learn a decision boundary that discriminate FG data. It's true, as in the LDA experiments, that the contrastive model does well in classification. The parameters of the generative contrastive model are also of interest; in the genomic study, they suggest novel biological states.

On projecting to a dense KxKxK tensor--that's exactly what we do for the HMM experiment, where K is small. We use the projection method similar to [9]. We've also tried the projection method on topics. It works well but we found it to be faster to exploit the sparsity in the moments. Multiple passes through the data can be avoided in the tensor power iteration by using orthogonal iteration which updates all K parameter vectors at once. The overall computation time is, up to a factor of ~2, the same as using multiple passes. Overall the running time is linear in input size. We'll include more discussion of this in the revision.

While it's true that the background (BG) can often be enormous, we want to include general settings to allow, for example, the BG to be sparse or to be a specific biological experiment and the FG to be larger. It's still interesting to understand the contrast in such cases. When the BG is large and complex, the efficiency advantage of our method is that we only need to compute its moments once instead of needing to learn a good model of it.

It's true that baseline 1 is also consistent. However, when the BG is sparse, we observe that standard LDA doesn't learn coherent topics, just subsets of keywords. It's not clear how to filter the shared topics in a principled way. It's also not clear how to extend baseline 1 to contrastive HMM. A full generative model capturing both FG and BG process is an interesting direction of research.